

# Brief Communication: Rejuvenating and strengthening the science-policy interface required to implement the Sendai Framework for Disaster Risk Reduction

Joel C. Gill[1, 2]

[1]Geology for Global Development, Cardiff, CF10 3AT, United Kingdom
[2]School of Earth and Environmental Sciences, Cardiff University, Cardiff, CF10 3AT, United Kingdom

*Correspondence to*: Joel C. Gill (joel@gfgd.org; gillj11@cardiff.ac.uk)

**Abstract.** Here we discuss the health of the science-policy interface required to support implementation of the Sendai Framework for Disaster Risk Reduction. Reflecting on the 2025 Global Platform for Disaster Reduction, we identify weaknesses in current mechanisms for scientific engagement. While the Sendai Framework highlights science as foundational to effective risk reduction, engagement remains limited by ad hoc structures and unclear processes. This article proposes three steps to revitalise the science-policy interface, emphasising inclusivity, synthesising scholarly contributions to support knowledge sharing, and dedicated thematic forums. Strengthening this science-policy interface is essential to realising the Sendai Framework's objectives through and beyond 2030.

## 1 Introduction

Endorsed by the United Nations General Assembly in 2015, the Sendai Framework for Disaster Risk Reduction aims to achieve the substantial reduction of disaster-related losses and risks across all sectors and scales (UNDRR, 2015). Central to this intergovernmental agreement is the recognition of science as a critical foundation for understanding, assessing, and mitigating disaster risk (Aitsi-Selmi et al., 2016). A decade on from its agreement, the global disaster risk reduction (DRR) community convened in Geneva, Switzerland, in June 2025 to assess progress, identify challenges, and propose strategies to accelerate implementation. This article reflects on that gathering, raises concerns about a potential weakening of the science–policy interface underpinning the Framework's goals, and considers the actions necessary to ensure the scientific community plays a full role in informing and shaping work to and beyond 2030.

Disasters are complex and interdisciplinary challenges, requiring contributions from diverse actors. While nation states have a primary responsibility to reduce disaster risk (UNDRR, 2015), the Sendai Framework encourages a multi-stakeholder approach, with a clear role for the science and technological community (Pearson and Pelling, 2015; Aitsi-Selmi et al., 2016). This includes those working in the natural, environmental, social, economic, health, and engineering disciplines (UNISDR, 2008), including a broad spectrum of geoscientists (e.g., geologists, seismologists, volcanologists, hydrologists, meteorologists, physical geographers, geomorphologists and others). The science and technology community are significant



contributors – alongside others – to understanding risk and its components, as well as designing and delivering effective risk reduction mechanisms (Gill and Bullough, 2017; Gill et al., 2020; Smith and Bricker, 2021).

For the work of natural hazard scientists to be useful, useable, and used in the context of risk reduction, a strong science-policy interface is required. Such interfaces are defined as '*social processes which encompass relations between scientists and other actors in the policy process, and which allow for exchanges, co-evolution, and joint construction of knowledge with the aim*

*of enriching decision-making'* (van der Hove, 2007). The aim of science-policy interfaces is to deliver decisions (both within and beyond the public policy domain) that are well-informed about the nature of the problem and the potential solution space, informed by the best available evidence (Van Enst et al., 2014). While they may be characterised as both a process or an organisation (Van Enst et al., 2014), typical shared requirements of an effective science-policy interface include (a) scientific networks engaging in a transparent manner, (b) genuine interdisciplinary interactions between social and natural sciences, and

(c) scientists exercising their responsibility as knowledge holders and technology developers (van der Hove, 2007).

In the following sections we look at how science is presented in the Sendai Framework and subsequent reporting and mechanisms for scientists to engage (**Section 2**), potential weaknesses in the existing science-policy interface supporting this Framework, as witnessed at the 2025 Global Platform for Disaster Risk Reduction (**Section 3**), and recommendations for strengthening this process (**Section 4**). Concluding remarks are set out in **Section 5**.

## 2 Science and the Sendai Framework

The Sendai Frameworks articulates a role for the scientific community in delivering its objectives, with several specific references to 'science' throughout:

- A guiding principle emphasises the need for "easily accessible, up-to-date, comprehensible, *science-based*, non-sensitive risk information, complemented by traditional knowledge" (UNDRR, 2015, Clause 19g, emphasis added).

- At a regional and global level, there is an agreed action to "enhance the *scientific and technical work* on disaster risk reduction and its mobilization through the coordination of existing networks and scientific research institutions at all levels and in all regions, with the support of the UNDRR Scientific and Technical Advisory Group" (UNDRR, 2015, Clause 25g, emphasis added).

- At national and local levels, agreed actions include supporting and facilitating *science-policy interfaces* for effective

decision-making in disaster risk management (UNDRR, 2015, Clauses 24h, 36b, emphasis added).

In this context, mobilisation of the scientific community is suggested to assist in enhancing methods and standards for risk assessments and disaster risk modelling, encourage effective data use, help identify gaps and priorities in research and technology, and support the integration of scientific knowledge into decision-making processes. (UNDRR, 2015). In the decade since the agreement of the Sendai Framework, science has been emphasised repeatedly to be instrumental in delivering

effective DRR. This was a key message of the mid-term review of the Sendai Framework, with the associated political declaration noting the "instrumental and cross-cutting role of science, technology and innovation in strengthening the



effectiveness and efficiency of disaster resilience-building" and encouraging more application of science to support and accelerate implementation of the Sendai Framework (United Nations, 2023, Clause 41). Taken together, these statements highlight a recognition that science is a central pillar to shaping and implementing effective DRR strategies. The UNDRR
Partnership and Stakeholder Engagement Strategy agrees and outlines some principal mechanisms by which scientists can engage with the Sendai Framework process (UNDRR, 2021), as summarised in **Table 1**.

**Table 1. Examples of mechanisms by which scientists can engage with the Sendai Framework process.**

| Engagement Mechanism | Membership / Leadership | Purpose | Further Reading |
|---|---|---|---|
| Global UNDRR Science and Technology Advisory Group | Closed/Limited Membership. This group consists of approximately 20 high-level experts. | Outlined in the Sendai Framework (UNDRR, 2015, Clause 25g), this group provides advice to UNDRR and the Special Representative of the UN Secretary General (the Head of UNDRR) on recent trends, challenges and opportunities for DRR. | UNDRR (2018); UNDRR (2021) |
| Regional UNDRR Science and Technology Advisory Groups | Closed/Limited Membership. A voluntary group of national and/or thematic experts (e.g., the European Scientific and Technical Advisory Group includes approximately 14 national experts and 8 thematic experts) | Support Sendai Framework implementation at regional and national levels, through scientific and technical advice to UNDRR and relevant countries. | UNDRR (2018); UNDRR (2021) |
| Science and Technology Partnership | Open to all. | Established as a broad, open network to strengthen the scientific and technical expertise for the implementation of clause 25(g). | UNDRR (2018); UNDRR (2021) |
| Major Group of Stakeholders: Scientific and Technological Community | Open to all. The Scientific and Technological Community Major Group (STC MG) is co-organised by the International Science Council and the World Federation of Engineering Organizations (WFEO). | Secures a mandate for science in UN forums and provides other stakeholders with an understanding of what is scientifically achievable. | UNDRR (2018); UNDRR (2021); United Nations (2025) |
| Bilateral Partnerships | Vary due to the nature of these partnerships. | Vary due to the nature of these partnerships. | UNDRR (2021); UNDRR (2025a) |
| | Example: Work with the *Integrated Research for Disaster Risk* (IRDR) programme of the International Science Council. | Example: IRDR's mission is to develop trans-disciplinary, multi-sectorial alliances for in-depth, practical DRR research, supporting the integration of research expertise from the sciences into policymaking to reduce disaster risk. | |

But while expectations of what scientists can offer to strengthen risk reduction are high (United Nations, 2023, Clause 41) and mechanisms for the community to engage are supposedly rich (see **Table 1**), evidence suggests that there is considerable scope



to rejuvenate and improve the structures and systems that facilitate dialogue with scientists. Of the different mechanisms listed in **Table 1**, several appear to be stagnant, lack clear guidance on how to participate, or are implemented in an ad hoc manner that hinders effective and inclusive participation. For example, at the time of writing, online information about the Global

UNDRR Science and Technology Advisory Group (STAG) includes a list of members from 2017–18 and terms of reference last updated in 2018. Online information about the European regional STAG lacks clarity regarding their terms of reference and when and how members are appointed. These challenges were not unforeseen. At the outset of the Sendai Framework implementation period, Carabine (2015) emphasised the need for the UNDRR STAG to be as open, inclusive and participatory as possible, highlighting concerns about the lack of clarity on how the STAG would be governed and structured. Furthermore,

at the time of writing, there is currently no online information about how to join the Science and Technology Partnership, and (as further outlined in **Section 3**) limited opportunities by the Scientific and Technological Community Major Group to support engagement in Sendai-related processes. While online information may not capture the full activity of a particular mechanism, it is used here as a measure of both accessibility and transparency, with sufficient cause for concern about the health of the mechanisms created to engage scientists.

**3 Reflections on the 2025 Global Platform for Disaster Risk Reduction**

The Global Platform for Disaster Risk Reduction is the UN General Assembly–recognised multi-stakeholder forum for reviewing progress on the implementation of the Sendai Framework, identifying gaps, and making recommendations to further accelerate action towards its stated objectives (UNDRR, 2025b). The 8th Session of the Global Platform took place in June 2025, co-organised by UNDRR and the Government of Switzerland.

As a multi-stakeholder forum, one would expect engagement by a wide range of stakeholders, including the science and technology community. The forum gathered more than 3600 people from 177 countries, with approximately 10% of these self-identifying as being part of a science and technology stakeholder group. If examining the convened sessions and oral and written statements by *other* stakeholder groups (e.g., parliamentarians), one can see a strong emphasis on the vital role of the scientific community in delivering the Sendai Framework. For example, the Government of the Philippines shared a report

from the *Asia Pacific Ministerial Conference on Disaster Risk Reduction* that noted the need to increase "science, technology and innovation, to help transform huge amounts of data into actionable information for local communities" (Government of the Philippines, 2025). However, in the run up to this event, there was no truly open coordination with the wider community by the Scientific and Technological Community Major Group, to feed into the full range of topics being discussed. During the Global Platform, there was no visible presence or reporting from the Global STAG and no statements delivered or placed

online by the Scientific and Technological Community Major Group coordinators. In contrast, civil society was well represented at the 2025 Global Platform, with participation from individual grassroots organisations, networks, and coordinating groups (e.g., the Global Network of Civil Society Organisations for Disaster Reduction, GNDR). The coordination among these was evident, inclusive and effective.





Individual scientists and scientific organisations did engage and contribute. For example, a statement was shared by Geology
105 for Global Development on behalf of the American Geophysical Union, European Geosciences Union, Geological Society of
London, International Union of Geodesy and Geophysics, Geology for Global Development, and Global Volcano Risk
Alliance. Some major scientific reports were also published during the Forum (e.g., the latest revisions to the UNDRR and
ISC (2025) Hazard Information Profiles, that involved liaising with a large and diverse group of scientists. These examples
differ from – but also demonstrate the huge benefits of – a coordinated, globally inclusive approach to policy engagement.
110 While there is a responsibility on Major Group coordinators and UNDRR (as the initiator of the STAG and Science and
Technology Partnership) to act, there is also a responsibility on scientific organisations, publishers, professional societies and
unions, as well as individual scientists to reflect on what more they can do to support a strong, effective science-policy
interface.

## 4 Recommendations to Strengthen the Science-Policy Interface

115 Given the importance of the scientific community to advancing the Sendai Framework (and DRR more broadly) and
recognising a potential weakening of the science-policy interface required to help facilitate engagement by and with scientists,
action is needed to reverse this. Here are three ideas (and seven recommendations) on how we can leverage the potential of
the scientific community to support global DRR efforts:

### 4.1 A refreshed, truly inclusive mechanism for the representation of science in the multi-stakeholder, multilateral
120 processes aligned with the Sendai Framework

The science and technology community can learn from other stakeholder groups (e.g., civil society) to develop and maintain
a structure that facilitates ongoing dialogue between the global science community and other DRR stakeholders. Examples of
specific recommendations on how to deliver this include:

 a. A clear and independent coordinating mechanism for the science and technology community, with readily accessible
125  information about this mechanism, ways of engaging, and the coordinating groups.

 b. Dedicated focal points (individuals or organisations) within international unions and other scientific organisations, to
  support a bidirectional flow of information into and out of that coordinating mechanism. The International Science
  Council could develop a set of 'focal point' role descriptions relating to multi-lateral processes (including, but not
  limited to, the Sendai Framework) that are distributed to member societies (e.g., the International Union of Geological
130  Sciences) to support them to establish these roles in their structures.

 c. A commitment to open and inclusive practice, ensuring meaningful opportunities to contribute to calls for evidence,
  the shaping of policy positions, and the design and delivery of activities at (for example) Regional and Global
  Platforms.



Coordinating organisations are resource-limited, and the proximity of the UN Ocean Conference and High-Level Political
Forum to the 2025 Global Platform may explain the limited engagement of the Scientific and Technological Community Major
Group in the latter. The recommendations above are not particularly, resource-intensive, and provide ways for major group
coordinators to leverage the expertise of others to support their work. **Figure 1** illustrates the approach set out in these
recommendations and illustrates how the work and resources indicated in **Section 4.2** could inform dialogue around DRR.

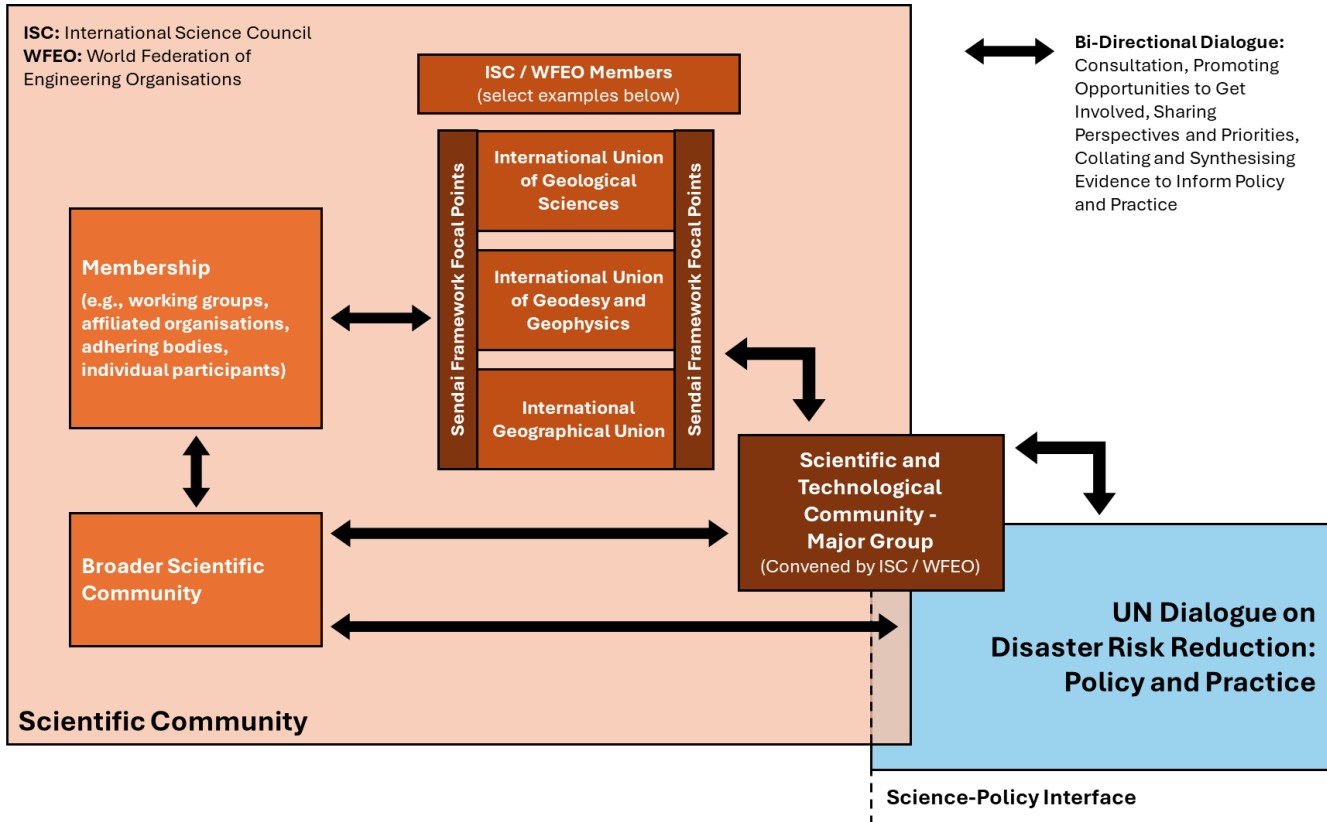


**Figure 1: A refreshed approach to increase participation in the Scientific and Technological Major Group to inform and support
implementation of the Sendai Framework for Disaster Risk Reduction.** In this adaptation of the current model, International Science
Council and World Federation of Engineering Organisation members are encouraged to nominate Sendai Framework Focal Points (either
individuals or organisations) to support the bi-directional flow of knowledge and strengthen the Science-Policy Interface. Direct engagement
and participation in Major Group activities by individual member organisations and the broader scientific community must also still be
possible.

## 4.2 Proactive collation and sharing of learning, case studies, priorities, and perspectives

There are many large gatherings of scientists around the world, with the results of millions of hours of work being presented.
These conferences, along with the scholarship captured in scientific journals and reports, represent a significant body of



evidence that can inform DRR actions at all scales. There's an unrealised potential to bring together and synthesise this learning into reports that support other stakeholders, such as those working in policy settings. Examples of specific recommendations on how to deliver this include:

    a.    Thematic (e.g., early warning, risk communication) and geographically specific (e.g., Central America, West Africa) reports capturing common learning presented at scientific conferences.

    b.    Major scientific journals with a risk reduction focus (e.g., Natural Hazards and Earth System Sciences) appointing a dedicated editorial role that focuses on promoting access to and understanding of the journal's content by other DRR stakeholders, to maximise impact and learning from scholarly work.

**4.3 A regular thematic platform focused on science, technology, and innovation, with the outcome document feeding**
**into the Global Platform**

Regional Platforms offer a potential model for focused discussion that is then integrated into the Global Platform. The annual UN Forum on Science, Technology, and Innovation for the SDGs, which feeds into the High-Level Political Forum on Sustainable Development, sets a welcome precedent for a thematic forum contributing to intergovernmental processes. A three-day Science and Technology Conference organised by UNDRR in 2016 was pivotal in mobilising the scientific community to
help implement the Sendai Framework. Examples of specific recommendations on how to deliver this include:

    a.    Convening a dedicated meeting to bring together the broad scientific and technological communities, as part of the range of events preceding and feeding into the shaping of a Global Platform (alternative options would be to have these feed into every other Global Platform, or to incorporate a Sendai focused day into the annual UN Forum on Science, Technology, and Innovation for the SDGs).

    b.    Preparing a formal outcome document, capturing the primary points of the thematic platform, for discussion at the Global Platform.

**Figure 2** sets out the approach set out in this recommendation and illustrates how the work and resources indicated in **Section 4.2** could be fed into the Global Platform dialogue.



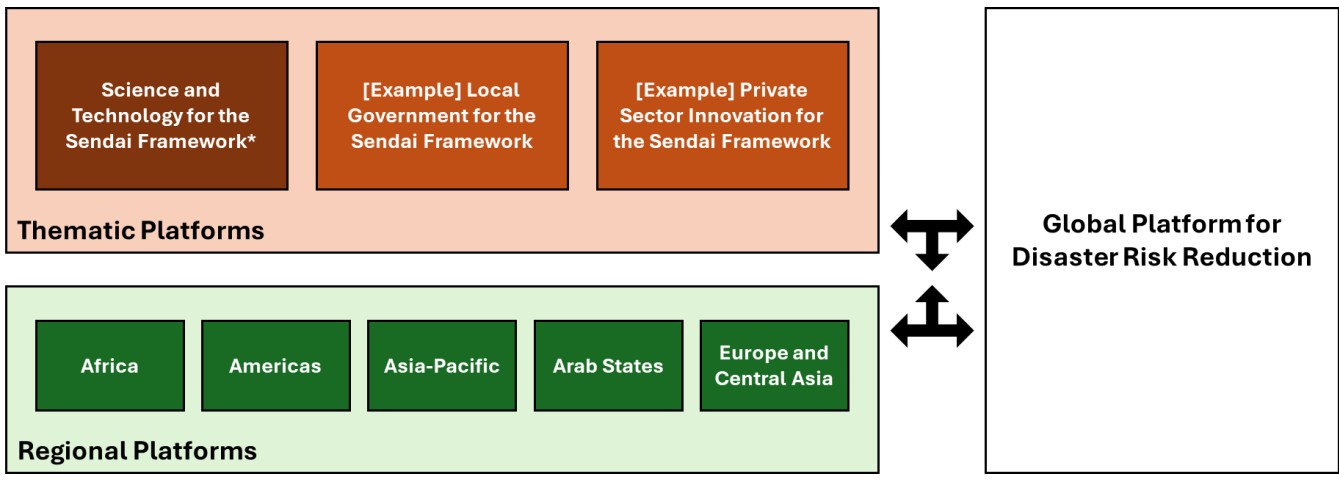

**\*Purpose:** (1) sharing perspectives and priorities, underpinned by evidence, to inform policy and practice, (2) strengthen the networks and mechanisms required for long-term engagement, and (3) hearing from other stakeholders to understand research and technological priorities to support Sendai Framework implementation.


**Figure 2: Thematic Platforms to inform the UNDRR Global Platform for Disaster Risk Reduction.** Modelled on existing Regional Platforms, and the success of the 2016 Science and Technology Conference on the implementation of the Sendai Framework for Disaster Risk Reduction 2015-2030, a set of thematic platforms could open up new opportunities to mobilise the scientific community and capture and share learning, case studies, priorities, and perspectives.

**5 Concluding Remarks**

While substantial mechanisms exist to support the science-policy interface required to support implementation of the Sendai Framework – work is needed to refresh these in the next 5-years as we seek to deliver action, and dialogues commence on the post-2030 agenda. The recommendations made in this article are not exhaustive but offer some initial perspectives on what can be done to improve inclusive engagement, strengthen the bi-directional flow of knowledge, and generate more impact

from scholarly work. Other actions will be needed, and the wider ecosystem of scientific organisations and individuals are encouraged to reflect on what they can offer.

The goal of the Sendai Framework is the substantial reduction of disaster risk and losses in lives, livelihoods and health and in the economic, physical, social, cultural and environmental assets of persons, businesses, communities and countries (UNDRR, 2015). Reducing disaster risk is key to advancing sustainable development objectives. As we work towards these

ambitions, we must recognise that structures and mechanisms can both hinder and catalyse progress. Achievements to-date can be lost or their full potential never realised, if we don't capture, share, and build on good practice and ensure improved access to scientific understanding, data, tools, and products. Rejuvenating and strengthening the science–policy interface required to deliver the Sendai Framework should be an urgent priority if we are to secure the progress expected—and needed—by communities around the world.



**Financial Support.** Attendance at the 2025 Global Platform for Disaster Risk Reduction was supported by the Lloyd's Register Foundation Grant TWRP\100012 (improving household preparedness in multi-hazard contexts). Work on this brief communication was supported by Cardiff University's Harmonised Impact Acceleration Account Strategic Impact Fund (H-IAA SIF). Geology for Global Development are supported financially by the International Union of Geological Sciences (IUGS).

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
