# Peer review of "Brief Communication: Rejuvenating and strengthening the sciencepolicy interface required to implement the Sendai Framework for Disaster Risk Reduction"

_EGUsphere, 2025_

## Author Comment (AC1)

**RESPONSE TO REVIEWERS AND SHORT COMMENTS (egusphere-2025-3559)**

**Brief Communication: Rejuvenating and strengthening the science-policy interface required to implement the Sendai Framework for Disaster Risk Reduction**

*Joel C. Gill*

The following comments are made in the context of this being a Brief Communication (and the scope of these being 2-4 pages, including those that *"report/discuss significant matters of policy and perspective related to the science of the journal, including "personal" commentary"*, with a maximum of 20 references - NHESS - Manuscript types).

**[RC1] Anonymous Referee #1**

*[RC1a] Thank you very much for your submission as a brief communication to the NHESS journal. I think the paper fits within the scope of the journal; however, I have some questions and remarks.*

[**Response to RC1a**] – Thank you for taking the time to provide a review for this work.

*[RC1b] Firstly, what I find missing is a clear problem statement and an explanation of why we need to re-think and re-start the science-policy interface for the implementation of the Sendai Framework. What is the actual problem? How does it work at the moment, and why does it not work currently? This remains very unclear in the current version. In addition, why are you mainly focusing on the scientific part while largely excluding the non-academic perspective?*

[**Response to RC1a**] Additional sentences will be added to the start of the introduction to ensure that the problem statement is clearer. In the context of reporting on the 2025 Global Platform, these sentences will highlight the **problem** that the flow of knowledge between the scientific community and policy-making community (already challenging) is weakening, and that this has implications on risk reduction. This is partly a **consequence** of ad hoc structures and unclear processes that do not ensure inclusive engagement in the science-policy interface at this level.

The reviewer questioned why I focused on *"the scientific part while largely excluding the non-academic perspective"*. The piece refers to 'science and technology' broadly, with the recommendations not being limited to academic engagement / perspectives (e.g., many international unions engage with scientists in industry and not just in academia, many in industry choose to publish research in scholarly journals). I recognise that people work in this space within and beyond academia. I will add sentences to **Section 1** that explicitly note the range of sectors and spaces by which the scientific community works in.

*[RC1c] Secondly, please reflect on your recommendations: why do we need these recommendations? What is new in them compared to what has already been known for more than a decade? Many of your points are already well known and have been discussed for a long time, both within and outside the scientific community.*

[**Response to RC1c**] – The reviewer indicates that these points are well known (for more than a decade) and have been discussed for a long time.

> There has been a weakening of the science-policy interface in the last decade; mechanisms set up and in place have become stagnant, are not transparent, and are not well connected to the wider science ecosystem. Issues may have been foreseen (and I refer to this in the text, with a referenced example), but the piece is bringing these issues now (10 years into Sendai) to light for a broader audience of scientists and particularly highlighting them in the context of a major intergovernmental meeting (June 2025) on disaster risk reduction. This context explains

why it is important to highlight these recommendations now, and their importance, even if they have been mentioned by others previously.

The reviewer suggests that the recommendations posed have been "*discussed for a long time, both within and outside the scientific community*" but does not evidence this with any suggested references to support these assertions. I do not see a rich body of literature (for example) discussing (i) the specific role of international unions in supporting a bidirectional flow of information into and out of an independent coordinating mechanism for the science and technology community, (ii) journal editorial roles focused on access to science by policy makers, or (iii) the longer-term value of a thematic science platform (alongside regional platforms), beyond papers in 2015-16 talking about the 2016 science platform (described in the article). Apologies if I have missed rich bodies of literature on these themes, I will do further checks to ensure no relevant literature is missed, and the reviewer is of course welcome to correct me and send some illustrative examples.

*[RC1d] Thirdly, why did you choose the concept of the science-policy interface without including more recent discourses? Why is the literature on transdisciplinary research, team science, and science-society interactions not included in your discussion? These areas provide a wide range of very interesting and useful concepts and ideas for changing the relationships and interactions between academia, policymakers, and non-state actors.*

[**Response to RC1d**] – I agree with the reviewer that there are a range of discourses relevant to the broader theme of the 'the science-policy interface required to implement the Sendai Framework for Disaster Risk Reduction'. The nature of a brief communication (vs a review article) means I don't have time to discuss everything and needed to put a clearer boundary to the scope of my reflections. I will address this comment by (i) pointing to the wider issues that should be considered, in the **Section 1 (Introduction),** and (ii) ensuring that the focus on this piece is more clearly articulated (mechanisms by which the scientific and technological community can feed into intergovernmental processes regarding disaster risk reduction).

*[RC1e] Fourthly, your recommendations are very broad and undefined. For example, regarding the point "a clear and independent coordinating mechanism, etc.": why and who would be responsible for it? How should it be organized and funded? Who provides the resources? Who has the capacity to conduct it? Who is liable for the actions? How do you ensure accountability and legitimacy? These aspects need to be addressed for each of your recommendations.*

[**Response to RC1e**] – I will look again at **Section 4 (Recommendations)** to see what additional detail can be added regarding the delivery of these recommendations. However, a purpose of the article is to provoke discussion about some of the questions raised in RC1e, and addressing these in a meaningful way is beyond the scope of a brief communication (2 to 4 pages). Most research papers provide recommendation for future work, policy changes or practice actions but don't include details of how these are funded or organised. For example, a paper recommending further research into x, y, z is not expected to set out what funding streams will be applied for to access that funding and how a consortium will be built and organised to ensure the project is effective. Identifying a potential 'next step' allows a wider group to discuss, explore, characterise in more depth, challenge and (if appropriate) implement that. Sentences will be added to **Section 1 (Introduction)** and **Section 5 (Concluding Remarks)** to ensure this purpose of the paper (for the community to consider how they respond to the challenge of a weakening science-policy interface) is clearer.

*[RC1f] Fifthly, please reconsider most of your suggested concepts. For example, learning: what do you mean by "learning" (single-, double-, triple-loop learning, or something else)? Regarding the "design and delivery of activities" on page 5: which activities, and how? Or the thematic groups on page 7, such as EWS or risk communication: what exactly should they do, and on which topics?*

[**Response to RC1f**] Thank you for highlighting where additional information is needed. I will review **Section 4** and ensure additional words are brought in to improve clarity. For example, the article (page 5) specifically says "design and delivery of activities at Regional or Global Platforms", a type of intergovernmental event described earlier in the piece. I will add clarifying words that note that typical activities at these Platforms include spoken interventions (on behalf of the stakeholder group), side events, and position papers released during the event. There is no reference to 'thematic groups' on Page 7 (or to risk communication / EWS), rather the article refers to a thematic platform (i.e., a dedicated science and technology conference talking place before an intergovernmental Global Platform, with an output document feeding into that process). I will look at Section 4 and see if additional clarity can be included, to help the reader.

*[RC1g] Lastly, please reflect on your figures. None of them are particularly helpful in clarifying what needs to be changed, what needs to be done, who should do it, or who is responsible for the transformation process.*

[**Response to RC1g**] Figures 1 and 2 (and their captions) will be looked at to consider how they could be adapted to improve clarity regarding the change process.

There is quite a lot of detail in Figure 1 regarding 'who' and I can better cross reference with relevant sections regarding how (e.g., Section 4.1, where a point is made about the International Science Council developing a set of 'focal point' role descriptions relating to multi-lateral processes and sharing with member societies).

With respect to Figure 2, I think this comment links to **Response to RC1e,** regarding what is reasonably expected of a brief communication in terms of detail of proposed mechanisms. I'm not it's necessary to articulate *who* should coordinate a thematic mechanism to write about the benefits there would be of having that event.

**[RC2] Anonymous Referee #2**

*[RC2a] The brief communication raises very important questions about the science-policy interface (SPI) in global DRR processes. Several examples are given of how this interface seems to be stagnating, and that interest groups other than scientists were more prominent during the last global platform meeting.*

[**Response to RC2a**] – Thank you for taking the time to provide a review for this work.

*[RC2b] The ideas and recommendations presented are relevant and should be seen as input to an important discussion on the topic. The recommendations under 4.2 and 4.3 are more logical and well presented, while those under 4.1 need more elaboration and clarification. For example, are there potential disadvantages of organising a top-level scientific organisation (4.1a), parallel to the policy one? It is stated that this recommendation is not resource-intensive (row 136) – a claim that needs to be justified. The various consultations proposed in Figure 1 would likely require significant resources. While Figure 1 identifies the different actors and bodies, it is difficult to follow how the proposed processes will take place. An alternative would be to exclude 4.1 and Figure 1 from the paper to give more focus to 4.2 and 4.3.*

[**Response to RC2b**] – I think the points made in Section 4.1 are important, as they are likely to be key to opening up the engagement process to a wider group of scientists than those able to attend thematic platform events or publish in scientific journals. I will work to elaborate on the points as follows (i) reflecting on the disadvantages of organising a top-level scientific organisation, while also recognising that this needs to sit alongside other bottom-up mechanisms, (ii) articulating how this recommendation is scalable and can be implemented in ways that reflect the level of resource available (with meaningful results) – this will replace the argument that the work is not resource

intensive, and (iii) additional information in Figure 1 and the caption about how proposed processes will take place.

*[RC2c] It is clear that the author's scope is on the global mechanisms of SPI, but it would also be interesting if the author could reflect on top-down versus bottom-up perspectives – whether the global level is an appropriate level for an effective SPI. What could be the reason for the stagnant global SPI? Could SPI be done at other levels and systematically brought to the global level? This would be partly in line with the recommendations under 4.2 and 4.3, as I understand them (to summarize and communicate research in new ways, etc.).*

[**Response to RC2c**] – Thank you for the constructive suggestion (and this links with **RC1d**). I agree with the reviewer that there are a range of scales by which Science-Policy Interface mechanisms can and should exist. In the article I have tried to highlight the importance of the global level mechanism in the run up to 2015 (supporting the development of the Sendai Framework), how that has changed since, and concerns over how that may impact on both implementation of Sendai and the design of any successor framework. As a brief communication I don't have time to discuss everything and needed to put a boundary to the scope of my reflections. However, the point raised is valid and I propose to add broader reflections to **Section 2 (Science and the Sendai Framework)** that talks to the need for a multi-level approach to SPIs.